# Flexible Thermoelectric Reduced Graphene Oxide/Ag_2_S/Methyl Cellulose Composite Film Prepared by Screen Printing Process

**DOI:** 10.3390/polym14245437

**Published:** 2022-12-12

**Authors:** Jianjun Wang, Yong Du, Jie Qin, Lei Wang, Qiufeng Meng, Zhenyu Li, Shirley Z. Shen

**Affiliations:** 1School of Materials Science and Engineering, Shanghai Institute of Technology, 100 Haiquan Road, Shanghai 201418, China; 2The Center of Functional Materials for Working Fluids of Oil and Gas Field, Sichuan Engineering Technology Research Center of Basalt Fiber Composites Development and Application, Southwest Petroleum University, Chengdu 610500, China; 3CSIRO Manufacturing, Private Bag 10, Clayton South, Melbourne 3169, Australia

**Keywords:** reduced graphene oxide, silver sulfide, methyl cellulose, thermoelectric composites, flexible

## Abstract

As an organic−inorganic thermoelectric composite material, a flexible, reduced graphene oxide (rGO)/silver sulfide (Ag_2_S)/methyl cellulose (MC) film was fabricated by a two-step method. Firstly, a rGO/Ag_2_S composite powder was prepared by a chemical synthesis method, and then, the rGO/Ag_2_S/MC composite film was prepared by a combined screen printing and annealing treatment process. The rGO and rGO/Ag_2_S composite powders were evenly dispersed in the rGO/Ag_2_S/MC composite films. A power factor of 115 μW m^−1^ K^−2^ at 520 K was acquired for the rGO/Ag_2_S/MC composite film, which is ~958 times higher than the power factor at 360 K (0.12 μW m^−1^ K^−2^), mainly due to the significant increase in the electrical conductivity of the composite film from 0.006 S/cm to 210.18 S/cm as the test temperature raised from 360 K to 520 K. The as-prepared rGO/Ag_2_S/MC composite film has a good flexibility, which shows a huge potential for the application of flexible, wearable electronics.

## 1. Introduction

One of the main issues human beings face in this century is the energy crisis. Traditional fossil energy, e.g., oil, coal, and natural gas, are still the main sources of energy used in the world. The consumption of the traditional fossil fuels inevitably produces solid waste and causes environmental pollution. Furthermore, the traditional fossil fuels are non-renewable, and, therefore, sustainable energy technologies have attracted more and more attention [1]. Thermoelectric (TE) materials can convert heat energy and electrical energy into each other [2,3]. TE devices have the virtues of being non-polluting and maintenance-free as well as having a long service life. TE devices show a great potential for relieving the energy crisis and environmental pollution [4]. So far, the TE generators have been used in many areas, such as aerospace [5], industry [6], biomedicine [7], and wearable electronics [8]. The low transformation efficiency of the TE generators is one of the main factors that limited their wide application. The materials’ TE properties, determined by the unitless figure of merit *ZT* (= *S^2^σT/κ,* where *T* is the absolute temperature and *S*, *σ*, and *κ* are the Seebeck coefficient, electrical conductivity, and thermal conductivity, respectively) [9,10,11], significantly influence the transformation efficiency of TE generators.

Compared with the traditional inorganic Bi−Te-based and Pb−Te-based alloys as TE materials, which contain toxic and rare elements [12], silver sulfide (Ag_2_S) is a potential TE material, due to its constituent elements that are non-toxic and naturally abundant [13]. Ag_2_S has three common forms: *α*—Ag_2_S (monoclinic), *β—*Ag_2_S (body-centered cubic), and *γ—*Ag_2_S (face-centered cubic) [14,15]. The *α*—Ag_2_S can transform to the *β—*Ag_2_S at ~450 K, and the *β—*Ag_2_S can turn to the *γ—*Ag_2_S at ~865 K [14,15]. For instance, Zhou et al. [16] fabricated an Ag_2_S bulk material by a spark plasma sintering method, and a *ZT* = 0.27 at 540 K was acquired. Duan et al. [17] prepared an Ag_1.96_S bulk material by a sintering process at a pressure of 2.5 GPa, and a *ZT* = 0.62 at 560 K was acquired. Wang et al. [18] fabricated an Ag_2_S ingot by a melting−annealing method, and a power factor (PF = *S^2^σ*) of 500 μW m^−1^ K^−2^ at 550 K was acquired.

The reduced graphene oxide (rGO) has the advantages of excellent electrical conductivity and mechanical properties. [19,20,21], which is always used as the filling phase for both inorganic and/or organic matrixes, and, therefore, shows a huge potential for the application of TE fields [22,23,24,25,26]. For example, Huang et al. [22] prepared a SnSe/rGO bulk composite through a spark plasma sintering process, and a *ZT* = 0.91 at 823 K was acquired for the bulk composite with a 0.3 wt% rGO. Gao et al. [23] prepared a tellurium nanowires/rGO film by a vacuum filtration method, and a PF = 80 μW m^−1^ K^−2^ at 313 K was acquired for the composite with a 50 wt% rGO. Mitra et al. [24] prepared a rGO/polyaniline composite by an in situ polymerization process, and a *ZT* = 0.0046 at room temperature (RT) was obtained for the composite with a 50 wt% rGO. Li et al. [25] fabricated a rGO/poly(3,4-ethylene-dioxythiophene):poly(4-styrenesulfonate) (PEDOT:PSS) composite via an in situ reducing process, and a PF = 32.6 μW m^−1^ K^−2^ at RT was obtained when the rGO content was 3 wt%.

The screen printing process has the virtues of economy, flexibility, strong adaptability, and easy operation; additionally, the thicknesses of the as-prepared materials can be adjusted in a wide range [27,28,29,30]. The screen printing process is always used for the preparation of polymer and inorganic/polymer TE materials [31,32,33,34,35,36]. For instance, in 2014, Wei et al. [31] prepared PEDOT:PSS films by a screen printing process on a paper substrate, and a PF = 34 μW m^−1^ K^−2^ at 473 K was achieved. In 2017, Shin et al. [32] prepared Bi_0.5_Sb_1.5_Te_3_ (p-type) and Bi_2_Te_2.7_Se_0.3_ (n-type) TE layers on fiber glass fabrics by a screen printing process using methyl cellulose (MC) as an additive. The printed layer was cured for 30 min at 250 °C–300 °C to solidify the sample and burn off the binders. After being hot pressed at 450 °C under 90 MPa for 5 min, a *ZT* = 0.65 (p-type) and *ZT* = 0.81 (n-type) were obtained for the TE layers at RT. In 2021, Liu et al. [33] fabricated an Ag_2_Se/polyvinyl pyrrolidone (PVP) composite film by a screen printing and sintering process on a polyimide substrate, and a PF = 4.3 μW m^−1^ K^−2^ at 390 K with the content ration of Ag_2_Se:PVP = 30:1 was achieved. In 2022, Amin et al. [34] prepared Bi_2_Te_3_ nanowires (NWs)/polyvinylidene fluoride (PVDF) composite films prepared by a combined screen printing and annealing process on a Kapton substrate, and a PF= 36 μW m^−1^ K^−2^ at 225 K with a 10 wt% PVDF was achieved. In 2021, our group [35] prepared flexible Bi_0.4_Sb_1.6_Te_3_/MC TE composite films on a mixed cellulose esters membrane by a screen printing process, and a PF= 2.32 μW m^−1^ K^−2^ at RT was achieved for the composite film with the volume fraction of 80% Bi_0.4_Sb_1.6_Te_3_. After being cold pressed, the PF enhanced to 10.07 μW m^−1^ K^−2^ at RT. In 2021, our group [36] also fabricated PEDOT:PSS/MC composite TE films on the PVDF substrate via a screen printing process, and a PF = 2.1 μW m^−1^ K^−2^ at 360 K was achieved for the composite film with the 25.67 wt% MC. After being treated using dimethyl sulfoxide, a PF = 16.2 μW m^−1^ K^−2^ at 340 K was obtained for the composite film. These research results show that MC was a good choice as the polymer matrix for the fabrication of the flexible TE composites via a screen printing process, and the screen printing process shows a huge potential for the fabrication of flexible TE materials.

Considering MC has a low thermal conductivity and good flexibility, Ag_2_S exhibits a high *S*, and rGO shows a high *σ*, in addition to the advantages of the screen printing process, the preparation of the ternary composite films, using the MC as the matrix and Ag_2_S and rGO as fillers via a screen printing process should deliver high TE properties. However, so far, few works about the flexible rGO/Ag_2_S/MC composite film have been reported. Herein, the rGO/Ag_2_S composite powder was prepared by a chemical synthesis method, and the rGO/Ag_2_S/MC composite film was prepared by a combined screen printing and annealing treatment process. The morphologies of the Ag_2_S powders and rGO/Ag_2_S/MC composite films as well as the TE properties of the rGO/Ag_2_S/MC composite films in the temperature range of 360 K to 520 K were studied.

## 2. Materials and Methods

### 2.1. Materials

Graphene oxide flakes (GO, size 0.5–5 μm) were bought from XFNANO Materials Tech. Co., Ltd. (Nanchang, China). A nylon membrane (diameter and pore size were 47 mm and 0.22 μm, respectively) was obtained from Millipore Co., Ltd. (Rockland, MA, USA). The sodium sulfide nonahydrate (Na_2_S·9H_2_O, 99%+), silver nitrate (AgNO_3_, ≥99%+), and methyl cellulose (MC, ≥ 99%+) were bought from Shanghai Titanchem Co., Ltd. (Shanghai, China).

### 2.2. Preparation of rGO/Ag_2_S Composite Powders

The GO flakes were ground into powders in the mortar and then added into the deionized water. After ultrasonication for 2 h, the Solution A was formed. An appropriate Na_2_S·9H_2_O was added in the Solution A with stirring for 3 h to form a Solution B. An appropriate AgNO_3_ was added to the Solution B with stirring for another 3 h, and the black precipitation was achieved after centrifuging and washing 3 times at 9000× *g* rpm for 5 min using deionized water. The rGO/Ag_2_S powders were finally obtained after drying at 70 °C for 12 h under a vacuum. The content of the rGO in the rGO/Ag_2_S powders was 0.02 wt%, which is the nominal composition. The rGO powders were prepared by the same process without adding the Na_2_S·9H_2_O and AgNO_3_.

### 2.3. Preparation of rGO/Ag_2_S/MC Composite TE Films

An amount of 0.1 g of the MC was added into 2 mL deionized water with stirring at 60 °C, and then 0.9 g rGO/Ag_2_S powders were added with stirring for 2 h to obtain the rGO/Ag_2_S/MC composite slurry. The rGO/Ag_2_S/MC composite film was prepared via a screen printing process, and the polyester screen mesh aperture was 200 mesh. The as-prepared film was dried at 70 °C for 12 h under a vacuum. The mass fraction of the rGO/Ag_2_S powders in the rGO/Ag_2_S/MC composite films was 90 wt%.

### 2.4. Post-Treatment of rGO/Ag_2_S/MC Composite TE Films

The rGO/Ag_2_S/MC composite film was cold pressed at 30 MPa for 5 min and then further annealed at 290 °C for 1 h under Ar protection. After cooling to RT, the rGO/Ag_2_S/MC composite film was achieved. Figure 1 shows the procedure for the fabrication and post-treatment of the rGO/Ag_2_S/MC composite TE film. Steps 1–2 show the preparation process of the rGO/Ag_2_S composite powders and rGO/Ag_2_S/MC composite films, respectively. Step 3 shows the post-treatment process of the rGO/Ag_2_S/MC composite films.

### 2.5. Characterization and Measurement

The morphologies of the Ag_2_S powders and rGO/Ag_2_S/MC composite TE films were observed by a scanning electron microscope (SEM, FEI Quanta 200 FEG, The Netherlands). The morphologies of the rGO/Ag_2_S powders were observed by the SEM (Zeiss Gemini 300, Germany). The morphologies of the Ag_2_S powders were observed by a transmission electron microscope (TEM, FEI Talos F200S, USA). The phase composition of the rGO/Ag_2_S/MC composite film was characterized by X-ray photoelectron spectroscopy (XPS) (Thermo Fisher Scientific ESCALAB 250Xi, USA). The *S* and *σ* of rGO/Ag_2_S/MC composite TE films were measured from 360 K to 520 K by an MRS-3 thin film TE test system (Wuhan Giant Instrument Technology Co., Ltd, China). In order to avoid the oxidation of the Cu contact electrode at a high temperature in the air, the samples were measured in a low vacuum atmosphere (≤40 Pa).

## 3. Results and Discussion

Figure 2a,b shows the SEM and TEM images of the Ag_2_S powders. The morphology of the as-prepared Ag_2_S powders was uniform, and the size of the Ag_2_S powders was ~80–200 nm. Figure 3a shows the XPS analysis of the C 1s spectrum of the GO and rGO powders. The C-O peak occurred at 286.6 eV as the GO was significantly reduced, indicating that the GO was reduced to rGO [37,38]. Figure 3b shows the SEM image of the rGO/Ag_2_S composite powers. It can be clearly seen that the rGO with the size of several micrometers exists in the rGO/Ag_2_S composite powers.

Figure 4a,b shows the SEM surface and fracture surface images of the rGO/Ag_2_S/MC composite TE films. It can be seen that after a combined cold-pressing and annealing treatment, the surface of the rGO/Ag_2_S/MC composite TE film was smooth, and some pores existed (see Figure 4a). The fracture surface image indicates the thickness of the rGO/Ag_2_S/MC composite film was intact and uniform, with an average thickness of 4.3 μm (see Figure 4b). Figure 4c–f shows the SEM image and corresponding SEM−EDS mapping of the rGO/Ag_2_S/MC composite film, which contains C, Ag, and S elements.

The TE performance of the rGO/Ag_2_S/MC composite film was tested at a variable temperature, and the results are shown in Figure 5. When the temperature ≤ 440 K, the *σ* of the rGO/Ag_2_S/MC composite film was < 0.20 S/cm, mainly because MC is an insulating polymer, and the *σ* of *α*—Ag_2_S is also very low near RT [39]. When the temperature increased to 480 K, the *σ* of the rGO/Ag_2_S/MC composite film reached a maximum value of 228.78 S/cm, mainly due to the transformation from *α*—Ag_2_S to *β*—Ag_2_S and the fact that *β*-Ag_2_S has a higher *σ* [40].

With the increased testing temperature from 360 K to 520 K, the absolute value of the Seebeck coefficient (|*S*|) of the rGO/Ag_2_S/MC composite film shows a huge change tendency. A maximum |*S*| was 439.74 μV/K at 360 K. When the temperature was ≤440 K, the |*S*| of the rGO/Ag_2_S/MC composite film was > 172 μV/K, and when the temperature increased to 480 K, the |*S*| significantly reduced to 66.50 μV/K; this might also be due to the transformation from *α*—Ag_2_S to *β*—Ag_2_S [41].

The TE performance of flexible materials is usually expressed by the PF. When the temperature < 440 K, the PF of the rGO/Ag_2_S/MC composite film was <0.5 μW m^−1^ K^−2^, mainly due to the low *σ* (< 0.2 S/cm), although it had a high |*S*| (> 172 μV/K). When the temperature > 440 K, the PF of the rGO/Ag_2_S/MC composite film increased significantly, and a maximum PF = 115 μW m^−1^ K^−2^ at 520 K was obtained. This value (115 μW m^−1^ K^−2^ at 520 K) is ~958 times higher than that of the PF at 360 K (0.12 μW m^−1^ K^−2^), mainly due to the significant increase in the *σ* of the composite film from 0.006 S/cm to 210.18 S/cm as the temperature rises from 360 K to 520 K. This value is also much higher than that of PEDOT:PSS films prepared by a screen printing process on a paper substrate (34 μW m^−1^ K^−2^ at 473 K [31]); a Ag_2_Se/PVP composite film prepared by a screen printing and sintering process on a polyimide substrate (4.3 μW m^−1^ K^−2^ at 390 K [33]); a Bi_2_Te_3_ NWs/PVDF composite film prepared by a combined screen printing and annealing process on a Kapton substrate (36 μW m^−1^ K^−2^ at 225 K) [34]; a Bi_0.4_Sb_1.6_Te_3_/MC TE composite film prepared by a combined screen printing and cold pressing treatment process on a mixed cellulose esters membrane substrate (10.07 μW m^−1^ K^−2^ at RT [35]); a PEDOT:PSS/MC composite TE film on the PVDF substrate prepared by a combined screen printing and dimethyl sulfoxide treatment process (16.2 μW m^−1^ K^−2^ at 340 K [36]); and a Bi_3.2_Sb_1.8_/Epoxy A composite thick film (14 μW m^−1^ K^−2^ at RT [42]). This value is much lower than that of the Bi_0.5_Sb_1.5_Te_3_ (p-type) and Bi_2_Te_2.7_Se_0.3_ (n-type) TE layers on flexible fiber glass fabrics prepared by a combined screen printing and hot-pressing process (2791 μW m^−1^ K^−2^ for p-type and 2077 μW m^−1^ K^−2^ for n-type at RT [32]), mainly due to the polymeric binders burning off and the Bi_0.5_Sb_1.5_Te_3_ and Bi_2_Te_2.7_Se_0.3_ TE layers being more dense after treatment with the hot-pressing method. Table 1 showed the *σ*, *|S|*, and PF of the rGO/Ag_2_S/MC composite TE film and those of previously reported TE composite materials.

Figure 6a shows the digital photo of the rGO/Ag_2_S/MC composite TE film. It can be seen that the rGO/Ag_2_S/MC composite TE film can be bent and cut into different configurations, indicating the rGO/Ag_2_S/MC composite TE film has a good flexibility. Figure 6b shows the rGO/Ag_2_S/MC composite film with a size of 1 × 2 cm^2^ can lift a weight of 270 g. This work further indicates that the screen printing technology can be used for the fabrication of cost-effective, flexible TE materials and generators [43], and, therefore, has a huge potential for the applications of wearable electronics.

## 4. Conclusions

A flexible rGO/Ag_2_S/MC thermoelectric film was prepared by a combined screen printing process and annealing treatment. The power factor of the rGO/Ag_2_S/MC composite TE film increased dramatically from 0.12 μW m^−1^ K^−2^ to 115 μW m^−1^ K^−2^ as the measured temperature increased from 360 K to 520 K, mainly due to the significant increase in the electrical conductivity from 0.006 S/cm to 210.18 S/cm of the composite film as the temperature increased. A maximum electrical conductivity, absolute value of the Seebeck coefficient, and power factor of 228.78 S/cm at 480 K, 439.74 μV/K at 360 K, and 115 μW m^−1^ K^−2^ at 520 K, respectively, was gained for the rGO/Ag_2_S/MC thermoelectric film. The as-prepared rGO/Ag_2_S/MC composite film shows good flexibility, which can be bent and cut into different configurations; therefore, it has a huge potential for the applications of flexible, wearable electronics.

## Figures and Tables

**Figure 1 polymers-14-05437-f001:**
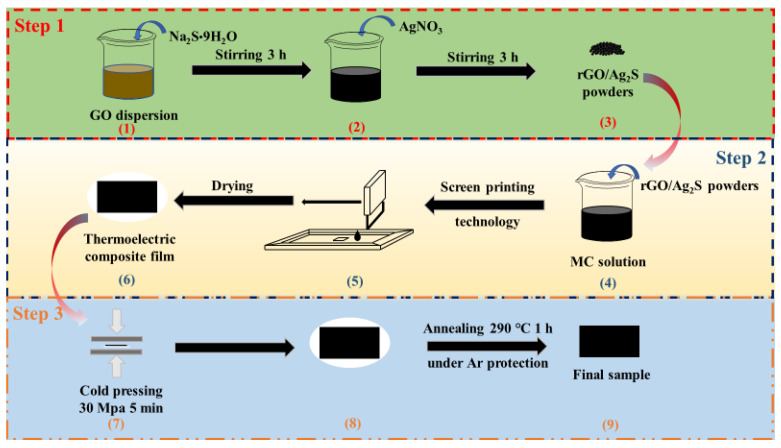
Schematic illustration of the fabrication and post-treatment process of the rGO/Ag_2_S/MC composite TE film. Step 1: preparation of the rGO/Ag_2_S composite powders. Step 2: fabrication of the rGO/Ag_2_S/MC composite TE films. Step 3: post-treatment of the rGO/Ag_2_S/MC composite TE films.

**Figure 2 polymers-14-05437-f002:**
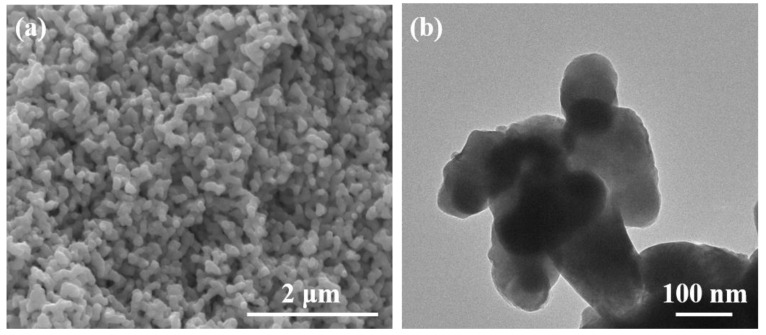
SEM and TEM images of the Ag_2_S powers (**a**,**b**).

**Figure 3 polymers-14-05437-f003:**
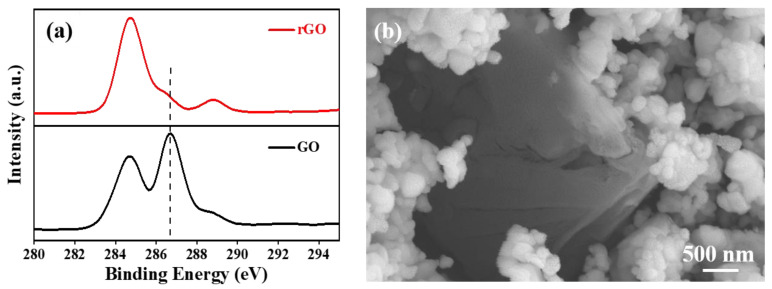
XPS analysis of the C 1s spectrum of the GO and rGO (**a**); SEM image of the rGO/Ag_2_S composite powers (**b**).

**Figure 4 polymers-14-05437-f004:**
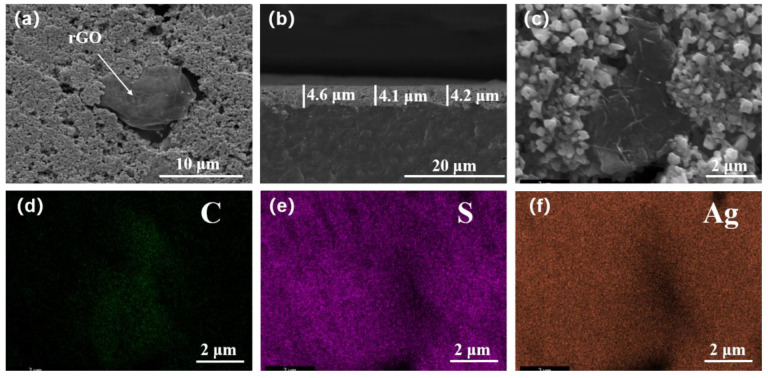
SEM surface image (**a**) and SEM fracture surface image (**b**) of the rGO/Ag_2_S/MC composite films; SEM image of the rGO/Ag_2_S/MC composite film (**c**); SEM−EDS mapping of the C element (**d**), S element (**e**), and Ag element (**f**) corresponding to (**c**).

**Figure 5 polymers-14-05437-f005:**
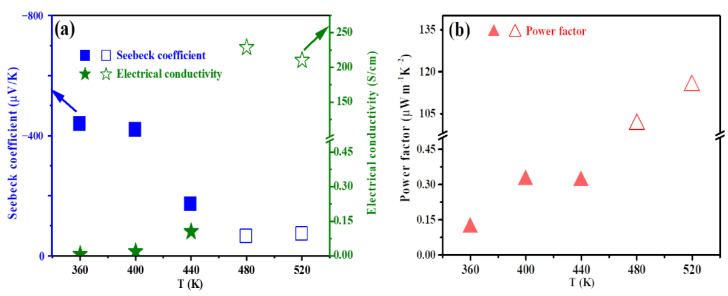
Temperature dependence of *σ* and *S* (**a**) and PF (**b**) of the rGO/Ag_2_S/MC composite film.

**Figure 6 polymers-14-05437-f006:**
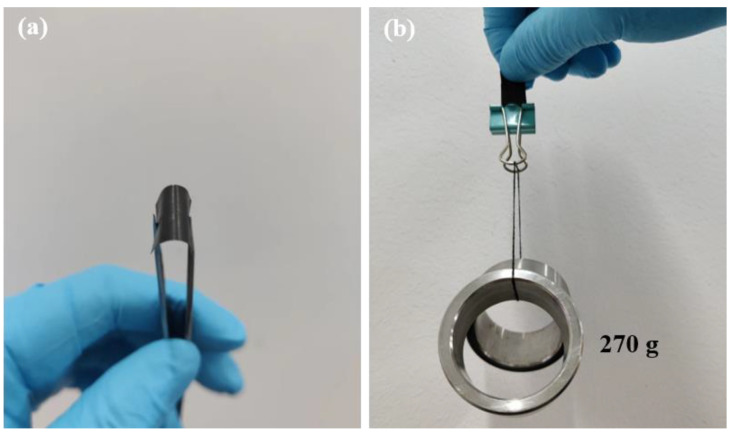
Digital photo of the rGO/Ag_2_S/MC composite film (**a**); the size of a 1 × 2 cm^2^ rGO/Ag_2_S/MC composite film can lift a weight of 270 g (**b**).

**Table 1 polymers-14-05437-t001:** The *σ*, *|S|*, and PF of the rGO/Ag_2_S/MC composite TE film and those of previously reported TE composite materials [31,32,33,34,35,36,42].

Author	Years	Methods	Post-Treatment	Materials	Type	*σ*^a^ (S/cm)	*|S|* (µV/K)	PF (µWm^−1^K^−2^)	Temperature	Reference
Wei et al.	2014	Screen printing		PEDOT: PSS film	P	550	25	34	473 K	[31]
Shin et al.	2017	Screen printing	Sintering and hot-pressing treatment	Bi_0.5_Sb_1.5_Te_3_	P	639	209	2791	RT	[32]
Shin et al.	2017	Screen printing	Sintering and hot-pressing treatment	Bi_2_Te_2.7_Se_0.3_	N	763	165	2077	RT	[32]
Liu et al.	2021	Screen printing	Sintering treatment	Ag_2_Se/PVP composites with the content ratio of Ag_2_Se:PVP = 30:1	N	~12.56	58.5	4.3	390 K	[33]
Amin et al.	2022	Screen printing	Annealing treatment	Bi_2_Te_3_ NWs/PVDF composite films with a 10 wt% PVDF	N	9.8	192	36	225 K	[34]
Niu et al.	2021	Screen printing	DMSO treatment	PEDOT:PSS/MC composite film with a 25.67 wt% MC	P	316.8	22.6	16.2	340 K	[36]
Li et al.	2021	Screen printing	Cold-pressing treatment	Bi_0.4_Sb_1.6_Te_3_/MC composite film with 80 vol.% of Bi_0.4_Sb_1.6_Te_3_ powders	P	4	158.5	10.07	RT	[35]
Cao et al.	2016	Screen printing	Annealing treatment	Bi_3.2_Sb_1.8_/ Epoxy A	N	~6.85	143.5	14	~RT	[42]
Wang et al.	2022	Screen printing	Cold-pressing and annealing treatment	rGO/Ag_2_S/MC composite film with 90 wt% rGO/Ag_2_S composite powders	N	210.18	73.96	115	520 K	This work

^a^ Some parameters were estimated according to the data in photograph or table in the References.

## Data Availability

The data are available on reasonable request from the corresponding author.

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
