# Peer review of "Flexible Thermoelectric Reduced Graphene Oxide/Ag2S/Methyl Cellulose Composite Film Prepared by Screen Printing Process"

_polymers, 2022, doi:10.3390/polym14245437_

Round 1
Reviewer 1 Report
This manuscript describes the organic-inorganic thermoelectric composite material was fabricated from rGO/silver sulfide (Ag2S) by chemical method and screen printing to create rGO/Ag2S/methyl cellulose (MC). The as-prepared rGO/Ag2S/MC composite film has good flexibility, which shows a huge potential for the application of flexible wearable electronics. The morphologies of Ag2S powders and rGO/Ag2S/MC composite film, as well as the TE 100 properties of the rGO/Ag2S/MC composite film were studied in the temperature range 101 from 440 K to 520 K. Although the authors have presented some information and results, I do not think that it is new and sufficient to prove the importance of this manuscript to justify publication in this journal due to for the following reasons:
1. This manuscript focuses on the synthesis of materials, however, the authors performed simple material analysis, which is not sufficient to demonstrate. Moreover, I didn't see any explanation for the material analysis content.
2. Authors should reanalyze more detail and clearer SEM, TEM, and XPS data. They are too rudimentary and don’t prove the fabricated material.
3. The flexibility of material can be measured by a universal testing machine with a specialized method instead of the only digital photo like Figure 6.
In my opinion, this manuscript should be submitted in other specialized journal after major revision. Therefore, I don’t recommend accepting this manuscript for publication in this journal.
Reviewer 2 Report
The manuscript has few of errors and unclear phrases or sentences also some unnecessary repetitions but which does not affect the quality of scientific work. The paper is interesting, but I find that at this moment seems to be not well referenced. Authors have not done a proper comparison of their approach with the state-of-the-art. The motivation of this research is not clear. There are several grammatical mistakes and inconsistencies in sentences. The authors have failed to clearly highlight the problem statement and the research gaps, for example by providing a thorough literature review and related works. What are the main contributions made by the paper? I am not so sure, how does the method differs from other state of the art methods? How does the performance compares with state of the art? Apart from some comparison on table 1, the comparative evidence is unclear. What were the criteria for selection of references to compare?
Reviewer 3 Report
The reviewed article presents the thermoelectric properties (Seebeck coefficient, electrical conductivity, power factor) of the three-component composite reduced Graphene Oxide/Ag2S/Methyl Cellulose made on flexible substrates using the screen printing technique. It is part of a wide range of searches for new thermoelectric materials, especially for flexible electronics applications. The presented issues are interesting and worth publishing, but after removing the shortcomings presented below:
1. Please provide more detailed information on how the authors made rGO/Ag2S powders containing only 0.02 wt% of reduced graphene oxide - Why such rGO content was chosen; with what accuracy the reduced Graphene Oxide was determinred in this system; whether the composites with a different rGO content were tested?
2. What screen density was used for the screen printing of three-component composite reduced Graphene Oxide/Ag2S/Methyl Cellulose? What material was used as the substrate?
3. Why the composite properties were measured at significantly reduced pressure (not at atmospheric pressure)?
4. The discussion on the thermoelectric properties of the tested composite is chaotic. I propose to see, for example, the review paper by M. Burton, G. Howells, J. Atoyo, M. Carnie; Printed thermoelectrics, Advanced Materials, vol. 34 (2022), art. ID 2108183 (44 pp.) and the paper by M. Gierczak, J. Prażmowska-Czajka, A. Dziedzic; Thermoelectric mixed thick-/thin film microgenerators based on constantan/silver, Materials, vol.11 (2018), art. ID 115 (9 pp.) and compare the thermoelectric properties of the tested composite with the ranges of values presented in these articles. This will allow to indicate whether the tested composite has a chance of being used in thermoelectric microgenerators. Of course, these articles should be included in the reference list.
Based on above mentioned remarks I propose revision of this manuscript and next review.
Round 2
Reviewer 1 Report
This manuscript describes the organic-inorganic thermoelectric composite material was fabricated from rGO/silver sulfide (Ag2S) by chemical method and screen printing to create rGO/Ag2S/methyl cellulose (MC). The as-prepared rGO/Ag2S/MC composite film has good flexibility, which shows a huge potential for the application of flexible wearable electronics. The morphologies of Ag2S powders and rGO/Ag2S/MC composite film, as well as the TE 100 properties of the rGO/Ag2S/MC composite film were studied in the temperature range 101 from 440 K to 520 K.
Although the original manuscript had many major problems, the author made positive and serious modifications to this study. It can be seen that the author has given the reasons in the material analysis to be clearer and added some data to prove it.
I think the manuscript has been greatly improved. Therefore, I recommend accepting this manuscript for publication in this journal.
Author Response
Thanks for the comments.
Reviewer 3 Report
The reviewed paper presents thermoelectric properties (Seebeck coefficient, electrical conductivity, power factor) of three-component composite reduced Graphene Oxide/Ag2S/ Methyl Cellulose made on flexible substrates with the aid of screen printing.
I would like to thank the authors for their explanation of my earlier doubts. But very few of these explanations are found in the revised version of the paper. Meanwhile, I believe that these explanation may be relevant also to other readers of this work.
That's why I propose:
1. Include in the article additional information from the answer from "The contentsof these rGO ..." together with Figure S8
2. Include in the article the answer to point 2 "What screen density was used ..? What material was used as the substrate?"
3. Include in the article the answer to point 3 "Why the composite properties were measured at significantly reduced pressure (not at atmospheric pressure)?"
Based on the above remarks, I propose minor revisions of this manuscript.
